# CapTalk: Text-Guided Stylization and Speech-Driven 3D Head Animation

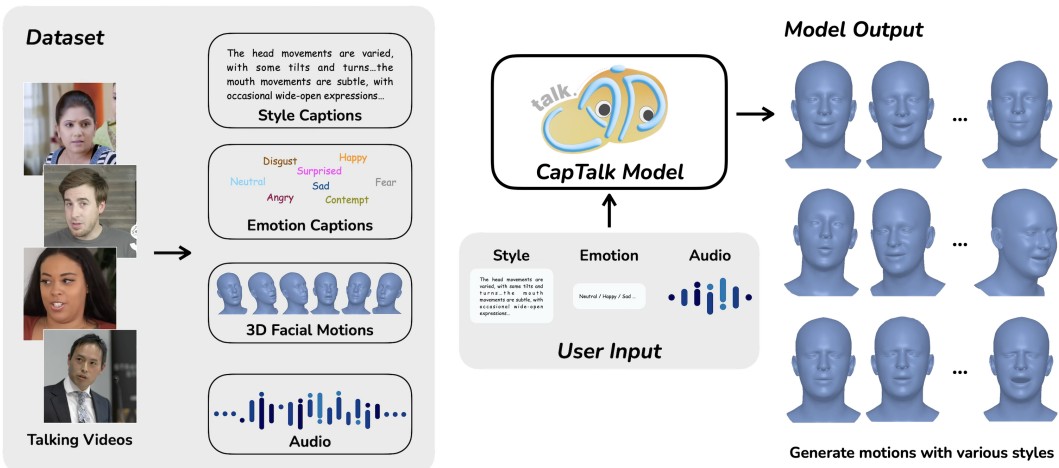

Figure 1: We present CapTalk, a framework generate 3D head motions from audio and text captions, enabling the real-time synthesis of realistic and stylized animation sequences. To achieve this, we constructed a new dataset with style and emotion captions.

## ABSTRACT

Audio-driven 3D facial animation aims to generate synchronized lip movements and expressive facial expressions from arbitrary audio inputs. However, existing methods typically rely on predefined identity or style latent features, restricting users' ability to flexibly control speaking styles. Moreover, applying a fixed style or identity throughout an entire audio segment often leads to facial animations that fail to adapt to the dynamic emotional content of speech. To overcome these limitations, we revisit the definition of speaking style and construct a large-scale dataset annotated with textual descriptions of both style and emotion. Building on this, we propose a novel talking head generation framework that enables fine-grained control over both speaking style and character emotion. Our model accepts textual descriptions of style and emotion alongside the driving audio, allowing real-time generation of highly synchronized lip movements and facial expressions that faithfully reflect the provided descriptions. Furthermore, our approach supports dynamic style and emotion control during inference, enabling the generation of facial animations that adapt to changing emotions within a single utterance. Experimental results demonstrate that our method achieves superior expressiveness and controllability compared to existing approaches.

## 1 INTRODUCTION

In recent years, the rapid advancement of large language models has driven significant progress in artificial intelligence, particularly in text generation and conversational interactions. However, many of these systems rely on text or speech for communication, often lacking the visual components needed for rich human-computer interaction. To bridge this gap and create more engaging user experiences, research on digital humans has attracted attention from both academia and industry.

Given the connection between speech and facial expressions, speech-driven 3D facial animation has emerged as a crucial element in crafting realistic virtual characters. Advancements in this field are therefore essential for expanding the use of digital humans in fields like education and entertainment.

The goal of speech-driven 3D head motion generation is to create realistic, synchronized facial expressions and head movements from audio input. This field has seen significant progress, with researchers exploring various generative models, such as autoregressive (Fan et al., 2022; Xing et al., 2023; Chu et al., 2025) and diffusion-based methods(Stan et al., 2023; Sun et al., 2024). While these techniques have achieved impressive lip synchronization and realistic facial motion, current methods for controlling the style of generated motions remain limited. Existing approaches (Fan et al., 2022; Xing et al., 2023; Peng et al., 2023a; Stan et al., 2023; Fan et al., 2024) often rely on fixed identity sets or utilize latent codes derived from specific identity motion sequences. This design restricts their ability to generalize to diverse or unseen speaking styles. Models trained on fixed identities struggle to scale beyond the identities in the training data, while those requiring motion sequences require additional videos, which is often cumbersome and impractical for end users.

To overcome the limitations of existing methods and enable style-controllable speech-driven motion generation, we introduce a new dataset with text-based annotations of speaking style, and then leverage this dataset to develop a model capable of text-guided speech-driven head motion generation. While speaking style is often associated with speaker identity, we observe that identity descriptions alone are insufficient to capture the speaking style. Instead, we define speaking style based on three key determinants: mouth movement amplitude, head movement amplitude, and emotion in this dataset. To annotate these features, we employ a combination of Vision-Language Models (VLMs) (Hurst et al., 2024; Team et al., 2024; Xu et al., 2025) and Audio-Language Models (ALMs) (Chu et al., 2023b; Wu et al., 2025). Since visual cues such as mouth and head movement amplitudes are readily extractable from video data, we employ a VLM to generate these annotations. In contrast, speech emotion is more accurately inferred from audio characteristics, prompting the use of ALM for emotion labeling. Based on these insights, we construct a dataset from YouTube videos. Each video segment is annotated with style annotations, emotion annotations, and frame-level FLAME head motion parameters.

Building upon the recent advances such as DiffPoseTalk and ARTalk, we propose a time-windowed, autoregressive speech-to-action model. The temporal receptive field afforded by time windows is essential for synthesizing high-quality, contextually coherent motions. For efficient and high-fidelity action generation, we adopt an autoregressive framework that operates both within and across time windows, enabling the capture of fine-grained facial expressions and the production of temporally continuous motion sequences. To integrate multi-modal information from both audio and text, we incorporate cross-modal fusion layers within the autoregressive process, aligning and fusing the precisely time-synchronized audio features and the more loosely text-based style captions. Furthermore, by leveraging historical action information, our model maintains overall motion continuity even when text-based style descriptions change between time windows.

The major contributions of our work are as follows:

- We introduce **CapTalk**, the first model to empower users with direct control over both the speaking style and emotion of generated motions via textual descriptions.
- We construct and will publicly release the first **large-scale 3D facial motion dataset** with rich annotations for speaking styles and emotions, to facilitate future research in this area.

## 2 RELATED WORK

### 2.1 SPEECH-DRIVEN HEAD MOTION GENERATION

Research on audio-driven 3D motion generation has been an active area for decades, with methodologies evolving substantially over time. Early approaches (Taylor et al., 2012; Xu et al., 2013; Edwards et al., 2016) primarily relied on procedural techniques, segmenting speech into phonemes and mapping them to predefined visemes using handcrafted rules. In recent years, learning-based methods (Fan et al., 2022; Xing et al., 2023; Lu et al., 2023; Aneja et al., 2024; Daněček et al., 2023; Peng et al., 2023b; Yang et al., 2024; Fan et al., 2024; Liu et al., 2024; Chae-Yeon et al., 2025; Wang et al., 2025; Chopin et al., 2025) have made significant advances, overcoming many limitations of

rule-based systems and enabling the generation of more natural and expressive facial animations. For instance, CodeTalker (Xing et al., 2023) introduces a model based on discrete motion priors, learning a codebook to map input audio to facial motion codes. EmoTalk (Peng et al., 2023c) presents an end-to-end framework for generating expressive 3D facial animations from speech and one-hot identity vector by disentangling emotional and content cues. FaceTalk (Aneja et al., 2024) leverages identity codes and audio features to generate facial motion within the expression space of 3D neural parametric head models. OT-Talk (Wang et al., 2025) employs optimal transport to synthesize facial motions from mesh and audio inputs, while Dimitra (Chopin et al., 2025) generates action latent vectors conditioned on audio and identity images. Despite these advancements, most existing methods utilize fixed identity and emotion vectors in some way, which limits their ability to generalize to a broader range of identities and speaking styles during inference. Additionally, some approaches (Ji et al., 2021; Sinha et al., 2022; Liang et al., 2022; Ji et al., 2022; Yi et al., 2022; Gan et al., 2023; Tan et al., 2024a; Zhang et al., 2023; Tan et al., 2024b; Hong et al., 2025; Zhen et al., 2025) focus on directly generating talking head videos instead of head motion. While effective for certain applications, this strategy restricts their integration with motion-driven downstream tasks, thereby limiting their broader applicability.

## 2.2 Stylized Speech-Driven Head Motion Generation

In recent years, stylized and emotionally expressive head motion generation has received increasing attention, aiming to create generative methods that are more expressive and can generalize to new identities. For example, EmoFace (Liu et al., 2024) introduces a dataset with emotions controlled via facial rig controllers, but it only provides emotion labels and is not publicly available. Diff-PoseTalk (Sun et al., 2024) and ARTalk (Chu et al., 2025) generate stylized facial animations using diffusion and autoregressive models, respectively, both guided by style embeddings extracted from reference videos. Similarly, ProbTalk3D (Wu et al., 2024a) proposes a non-deterministic, two-stage model that synthesizes facial animations conditioned on a style vector. These style feature-based approaches require users to supply reference videos and extract motion sequences to obtain style representations, which can be cumbersome when generalizing to new identities and styles. ModelSeeModelDo (Pan et al., 2025) introduces a style basis to guide a latent diffusion model, leveraging key poses from a reference video to ensure accurate style transfer; however, this method also necessitates user-provided template sequences. Among existing works, MEDTalk (Liu et al., 2025) is most closely related to our approach. It proposes to jointly generate facial motions using multiple modalities, including reference images, appearance descriptions, expression labels, audio, and audio text. However, it focuses more on generating emotional expressions, rather than encompassing broader aspects such as head movements. Furthermore, this work constructs text annotations and MetaHuman-based motions from a laboratory recordings dataset (1.5 hours), whereas we collect from in the wild videos (more than 200 hours) to achieve robust generation. In contrast to these methods, our model enables convenient and flexible control over generated motions by leveraging textual descriptions of speaking style, eliminating the need for reference videos or template sequences.

## 3 Dataset

Generating speech-driven head motions requires not only synthesizing synchronized facial expressions but also capturing the unique speaking style. Existing datasets often address this by collecting speech-driven facial motions linked to specific subject IDs, enabling personalized generation. However, this approach presents two main limitations: (1) identity-based datasets require extensive data from participants; (2) models trained on such datasets lack the ability to generalize to unseen identities. Alternatively, some datasets leverage in-the-wild videos to capture a broader range of identities and speaking styles, but it still difficult for users to specify desired speaking style at inference. To enable flexible, user-friendly style control, we introduce CapTalkingHead, a new dataset that uses natural language annotations to describe speaking style and emotion.

A key contribution of our dataset is the inclusion of style captions extracted from both audio and video streams, making it the first large-scale resource to provide multi-modal style supervision for the talking head generation task. Audio and video capture different but complementary aspects of speaking style: while audio reveals prosody, intensity, and emotional tone, the visual modality

conveys head motion, lip dynamics, identity-related traits, and visible affect. By combining the two, we obtain a holistic description of speaking style that cannot be captured by either modality alone.

For audio, we employ the Qwen-Audio-Chat model (Chu et al., 2023a), prompted to output an `emotional_label` chosen from the predefined set {angry, disgust, contempt, fear, happy, sad, surprised, neutral}. For video, we use a finetuned Qwen2.5-VL 7B model (Bai et al., 2025), adapted to ignore background context and instead focus on human appearance (mainly head shape), mouth opening size, and head movement amplitude during speaking. The combined use of these two captioning pipelines provides comprehensive style annotations, enables fine-grained controllability in talking head generation, and fully utilizes the audio and video information in the original data.

After collecting and processing, our dataset comprises 24,441 video clips, totaling approximately 200 hours. All videos are standardized to 25 frames per second and 16,000 Hz audio, resulting in 18,074,445 frames, with an average clip duration of approximately 29 seconds. Each video clip is paired with a style description derived from the video content, an emotion annotation extracted from the audio, and corresponding FLAME motion parameters.

## 4 METHOD

We provide an overview of our method in Figure 2. We first train a codec model on the FLAME parameters, thereby achieving a discrete representation of the motion space. We then train an autoregressive model guided by speech and text captions to generate motion code in the discrete space. In the following sections, Section 4.1 details the problem definitions, Section 4.2 explains the multi-scale codec model, and Section 4.3 introduces the speech and caption-guided autoregressive model.

### 4.1 PRELIMINARIES

We adopt the widely used 3D deformable model (3DMM) FLAME (Li et al., 2017) to represent facial motion, where motion is modeled by shape $\beta$, expression $\psi$, and pose $\theta$. Given these parameters, a face mesh including 5,023 vertices $V$ can be reconstructed using blendshapes and rotation operations. And we define the motion vector $M$ as the concatenation of the $\psi$ and $\theta$ over N frames.

### 4.2 MULTI-SCALE CODEC

Predicting motion frames from speech is a challenging task due to the dense temporal structure and complex mapping between modalities. Drawing inspiration from the success of discrete representations in image and motion generation (Xing et al., 2023; van den Oord et al., 2017; Zhou et al., 2022; Chu et al., 2025) and the effectiveness of multi-scale modeling (Jung et al., 2024; Tian et al., 2024; Chu et al., 2025), we adopt a multi-scale binary quantized codec to efficiently capture motion dynamics.

Given an input motion sequence of $N$ frames, we first employ a transformer encoder to process features and project them into a latent space. We then quantize the latent representation $L_n$ into binary, multi-scale discrete codes $C_{lvl}$ using binary spherical quantization (Zhao et al., 2024). $C_{lvl}$ of different levels are obtained by resizing the original latent sequence $L_n$ to the corresponding length in the temporal dimension and then quantize it. And the residuals are then subtracted from each scale during the quantization process. During the decoding process, the discrete codes $C_{lvl}$ at each scale are upsampled to length $n$ and then summed. A transformer decoder then reconstructs the motion sequence $\hat{M}_n$ from these summed codes. The overall process is illustrated in Figure 2 (a).

To train the VQ autoencoder, we employ a hybrid loss function that balances motion accuracy and codebook stability as follows:

$$L_{\text{Codec}} = \|\hat{M}_n - M_n\|_1 + w_{\text{full}}\|\hat{V} - V\|^2 + w_{\text{lips}}\|\hat{V}_{\text{lips}} - V_{\text{lips}}\|^2 + L_{\text{vq}}, \tag{1}$$

where $M_n$ and $\hat{M}_n$ denote the FLAME motion sequences, $V$ and $V_{lips}$ represent the full head mesh vertices and lip region vertices, and $L_{\text{vq}}$ is the loss for binary code stability. By jointly optimizing these objectives, our model learns a compact and expressive motion representation that models local and long-range dependencies in discrete space, ensuring high-fidelity synthesis and strong temporal consistency.

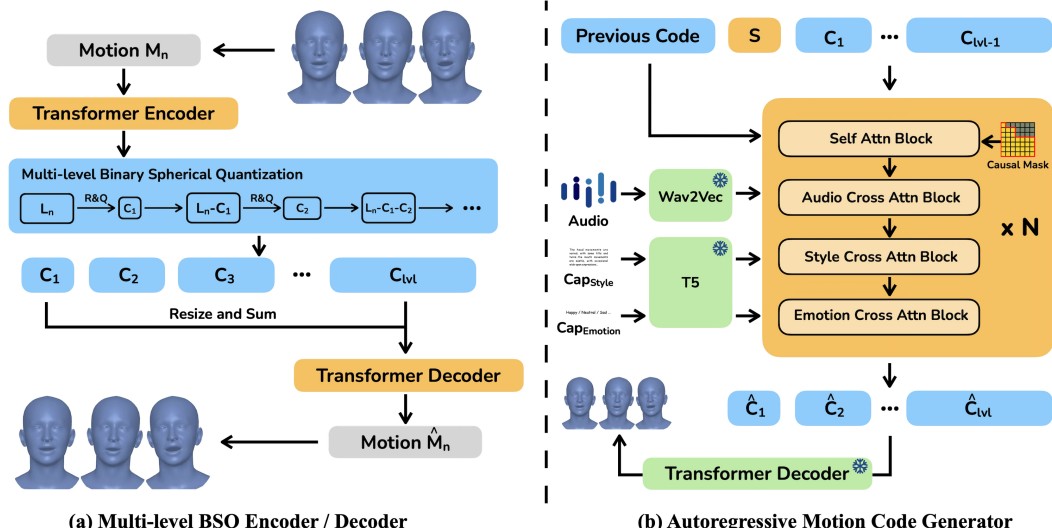

**(a) Multi-level BSQ Encoder / Decoder**   **(b) Autoregressive Motion Code Generator**

Figure 2: Figure (a) shows our multi-scale codec. It encodes motion $M_n$ into binary codes $C_1$, $C_2$, ..., $C_{lvl}$ of different lengths. The motion latent is resized to the length of $C_i$ and residually quantized to get $C_i$. We then resize $C_i$ to length $n$ and sum them for decoding. Figure (b) shows our autoregressive generator. Based on the previous window codes, previous scale codes $C_{1,i-1}$, audio features, style and emotion text features, the next scale code $C_i$ is generated through autoregression. After completing the current window, the generation of the next window begins with start token $S$.

### 4.3 Speech-to-Motion Autoregressive Model with Captions

After training the encoder-decoder, we obtain discrete motion encodings. To achieve long-term temporal consistency and high-quality synthesis, we autoregressively model these encodings using a Transformer that operates across both temporal windows and code scales. For speech feature extraction, we employ a multilingual pre-trained wav2vec2 model (Baevski et al., 2020). Style and emotion features are extracted using a T5 model (Raffel et al., 2020a), with style captions derived from video content and emotion captions from audio. We process information from different modalities by stacking self-attention layers and cross-attention layers in the model. For audio feature injection, we utilize a carefully designed Rotary Position Embedding (RoPE) (Su et al., 2024) to ensure precise temporal alignment between each code at every scale and its corresponding audio feature, thereby achieving time-synchronized audio conditioning. In contrast, for time-insensitive textual style control, we do not apply positional alignment between textual feature and the code features. Instead, we retain only the position encoding within the code features, allowing the textual control information to influence all code levels within a window uniformly. The overall process is shown in Figure 2 (b). We supervise the model using a cross-entropy loss and introduce label perturbations during training to enhance the diversity and robustness of the generation process.

## 5 Experiments

In this section, we first introduce the details of the dataset, provide an overview of the implementation of our method, describe the metrics used, and present the baseline methods. Subsequently, we compare our method with existing approaches across a range of evaluation metrics.

### 5.1 Experiment Setting

**Datasets.** We introduce CapTalkingHead, a novel dataset consisting of 24,441 video clips, 18,074,445 motion frames, and a total length of 200.8 hours. We reconstruct FLAME parameters of each frames and provide textual annotations describing the speaking style and emotion of each video. Further details of the dataset are provided in Section 3 and Appendix C. To assess the

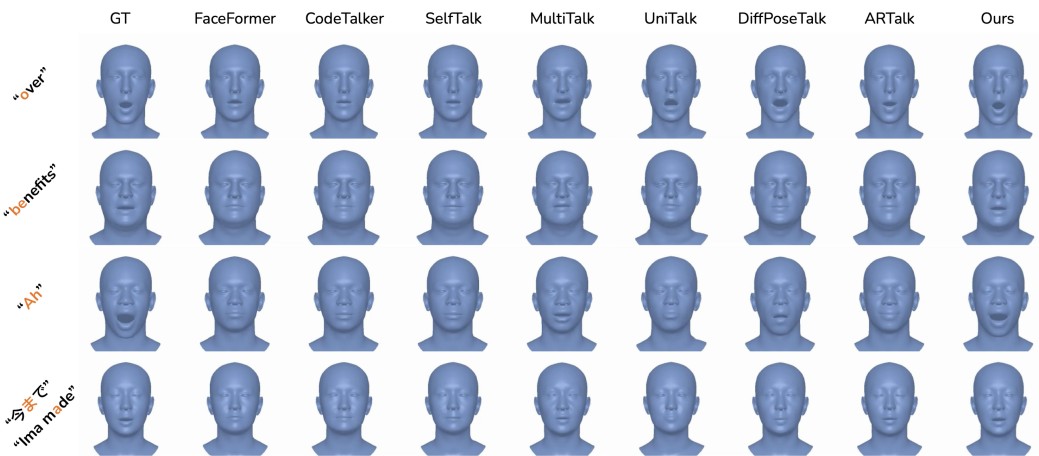

Figure 3: Qualitative comparison with existing methods (all head poses fixed). Our method shows better alignment with the ground truth in expression style, mouth dynamics, and lip synchronization. Additional videos results are available in the supplementary materials.

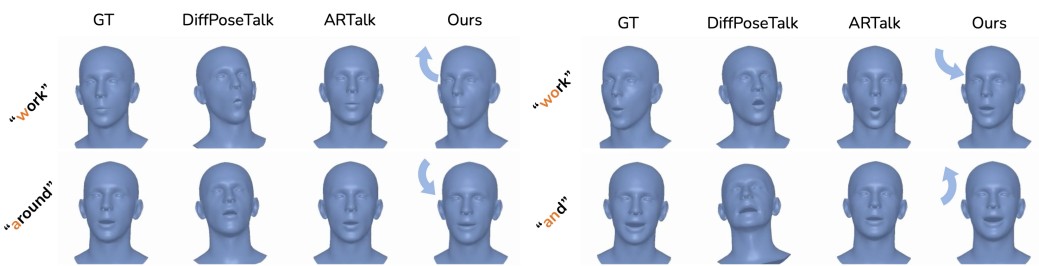

...she walked around the neighborhood and didn't see...

Style caption: "The person in the video has short, dark hair and is wearing a gray shirt. They are speaking with a neutral to slightly animated expression, with their mouth moving in a way that suggests they are speaking clearly and with some emphasis. The head movements are moderate, with occasional turns and tilts, indicating engagement with the audience or a specific point being made. The overall speaking style appears to be natural and expressive, with a slight amount of animation to convey enthusiasm or emphasis on certain points." Emotion caption: "happy"

Figure 4: Qualitative results of head pose. When certain words are stressed, our method generates head movements that are similar to human behavior and consistent with the style text description.

generalization capability of our model, we also conduct evaluations on the test split of the widely used MEAD dataset (Wang et al., 2020), which contains expression annotations but only provides video data. We apply the same tracking pipeline to extract the corresponding FLAME parameters from MEAD. It is worth noting that at inference time, we only utilize the original MEAD emotion labels as emotion captions, and do not extract or provide additional style captions.

**Evaluation Metrics.** Based on previous studies (Richard et al., 2021; Fan et al., 2022; Xing et al., 2023; Nocentini et al., 2024; Sun et al., 2024), we adopted the lip vertex error (LVE) (Richard et al., 2021) and the upper face dynamic deviation (FDD) (Xing et al., 2023) to evaluate the generated facial motions. LVE calculates the maximum error of lip vertices for each frame, evaluating the largest error between predictions and ground truth. FDD calculates the standard deviation of the motion of each upper facial vertex over time between predictions and ground truth, evaluating the consistency of upper facial motion, which is closely related to speaking styles. In addition, we use the mean head distance (MHD) to measure the average difference of full head vertices. Compared to LVE, MHD is less sensitive to temporal synchronization and includes some evaluation of expressions. We also use lip open dynamic deviation (LODD) and head pose dynamic deviation (HPDD) to assess the quality of style control. LODD measures the standard deviation of the upper and lower lip distance over time between the predicted and ground truth motions, assessing mouth opening size, which is closely related to speaking style. HPDD measures the standard deviation of the head rotation between the predicted and ground truth motions, assessing the control of head movement.

Table 1: Quantitative results on the CapTalkingHead test split. We use colors to denote the first and second places respectively. These results marked with * are from methods that cannot generate head poses, and we calculated them with all zero head poses.

| Method | LVE ↓ | MHD ↓ | FFD ↓ | LODD ↓ | HPDD ↓ |
|---|---|---|---|---|---|
| FaceFormer (Fan et al., 2022) | 13.24 | 3.03 | 37.00 | 205.81 | 13.13* |
| CodeTalker (Xing et al., 2023) | 12.55 | 2.83 | 38.93 | 170.52 | 13.13* |
| SelfTalk (Peng et al., 2023a) | 12.46 | 2.81 | 36.16 | 175.84 | 13.13* |
| MultiTalk (Sung-Bin et al., 2024) | 12.13 | 2.72 | 34.59 | 110.30 | 13.13* |
| UniTalker (Fan et al., 2024) | 13.68 | 3.39 | 33.91 | 132.27 | 13.13* |
| DiffPoseTalk (Sun et al., 2024) | 11.38 | 2.50 | 29.36 | 82.30 | 9.59 |
| ARTalk (Chu et al., 2025) | 7.71 | 1.98 | 29.64 | 90.89 | 9.76 |
| CapTalk (Ours) | 6.44 | 1.80 | 25.14 | 58.27 | 7.59 |

Table 2: Quantitative results on the MEAD (Wang et al., 2020) test split. We use colors to denote the first and second places respectively. It is worth noting that our method is **not** trained or fine-tuned on MEAD. Furthermore, compared to DiffPoseTalk and ARTalk, which use reference action sequences, our method only utilizes emotion labels from the original data, meaning we can only capture very few style cues.

| Method | LVE ↓ | MHD ↓ | FFD ↓ | LODD ↓ |
|---|---|---|---|---|
| FaceFormer (Fan et al., 2022) | 15.60 | 3.45 | 25.21 | 243.06 |
| CodeTalker (Xing et al., 2023) | 13.95 | 3.09 | 27.62 | 200.85 |
| SelfTalk (Peng et al., 2023a) | 14.26 | 3.05 | 27.32 | 230.96 |
| MultiTalk (Sung-Bin et al., 2024) | 12.89 | 2.84 | 27.82 | 100.55 |
| UniTalker (Fan et al., 2024) | 16.13 | 3.86 | 27.52 | 173.31 |
| DiffPoseTalk (Sun et al., 2024) | 10.19 | 2.44 | 23.33 | 79.99 |
| ARTalk (Chu et al., 2025) | 8.12 | 1.73 | 18.51 | 109.80 |
| CapTalk (Ours) | 8.07 | 1.81 | 20.59 | 85.82 |

**Baseline Methods.** We conduct a comprehensive evaluation of our method against leading academic baselines across two datasets: CapTalkingHead and MEAD (Wang et al., 2020). The baseline methods FaceFormer (Fan et al., 2022), CodeTalker (Xing et al., 2023), and SelfTalk (Peng et al., 2023a) are mesh-based methods, where we input the corresponding mesh and specify its first speaker identity for inference. For MultiTalk (Sung-Bin et al., 2024), which supports language-based stylization, we used its English style for evaluation. For UniTalker (Fan et al., 2024), we compute the metrics using the meshes generated by the generalized pivot identities. For DiffPoseTalk (Sun et al., 2024) and ARTalk (Chu et al., 2025), we use pre-trained weights for evaluation and input the first few seconds of the ground truth motion clip as style reference.

## 5.2 QUANTITATIVE RESULTS

We present the quantitative comparison on CapTalkingHead test split in Table 1. The results show that our method achieving significant improvements in lip synchronization accuracy (LVE) and outperforms the baseline in style control (MHD, FFD, LODD, HPDD), indicating that our method not only achieves precise lip synchronization but also effectively captures the specified speaking styles. We also present quantitative comparison results on the MEAD test set in Table 2 . Because the head in the MEAD dataset is rarely motionless, we do not evaluate head pose-related metrics. Furthermore, our method only utilizes emotion labels, not style description labels as input. This means that our method can obtain very few style cues. However, compared to baseline methods, especially DiffPoseTalk and ARTalk, which use ground truth action sequences as reference action sequences, our method still demonstrates competitive performance.

Table 3: User study results. The percentages represent **the proportion of users who preferred the our method over the baseline** in each category. Among them, Sync represents lip synchronization. Style, Expression and Pose represent the consistency and naturalness of style, facial expression and head pose respectively.

| Method | Sync (%) | Style (%) | Expression (%) | Pose (%) |
|---|---|---|---|---|
| vs FaceFormer (Fan et al., 2022) | 0.63 | 0.75 | 0.73 | - |
| vs CodeTalker (Xing et al., 2023) | 0.83 | 0.77 | 0.87 | - |
| vs SelfTalk (Peng et al., 2023a) | 0.92 | 0.96 | 0.98 | - |
| vs MultiTalk (Sung-Bin et al., 2024) | 0.94 | 0.94 | 0.88 | - |
| vs UniTalker (Fan et al., 2024) | 0.96 | 0.92 | 0.94 | - |
| vs DiffPoseTalk (Sun et al., 2024) | 0.79 | 0.78 | 0.79 | 0.72 |
| vs ARTalk (Chu et al., 2025) | 0.68 | 0.66 | 0.68 | 0.67 |

## 5.3 QUALITATIVE RESULTS

In Figure 3, we present a qualitative comparison between our method and other baseline approaches. Our method demonstrates excellent lip synchronization, accurately capturing various speech elements. Furthermore, the generated facial expressions and mouth openings are close to ground truth, demonstrating our excellent style control capabilities. In Figure 4, we demonstrate the capabilities of our approach with respect to head movements. The results show that our approach can well capture accents and generate plausible head movements. In Figure 5, we show results for different styles and emotions inputs using the same audio input. This demonstrates that our method is able to respond to changes in text input while feeding the same input speech. For more dynamic qualitative evaluation results, please refer to the **supplementary videos**.

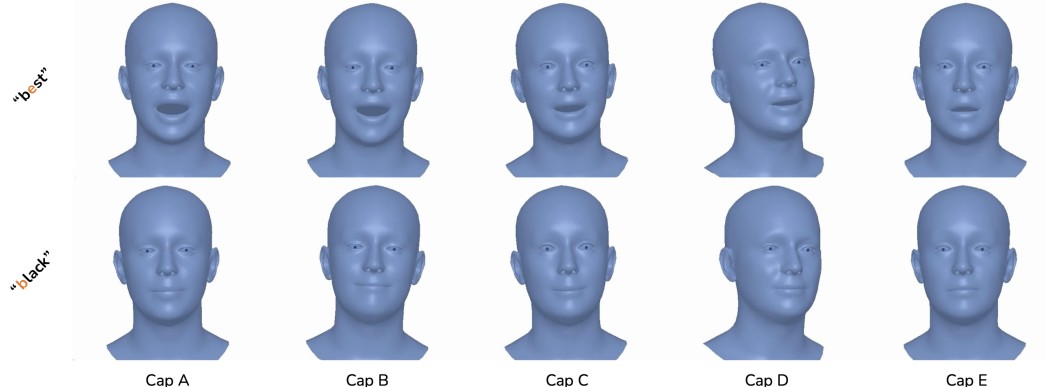

Figure 5: Qualitative results for style control. We fixed the input speech and varied only the text input. The results show that our method generates facial motions that correspond to the input style and emotion captions. For details on Caps A, B, C, D, and E, please refer to the Appendix F. Video results are also provided in the supplementary material.

## 5.4 USER STUDY

User studies are a reliable method for evaluating generation models. To comprehensively compare our method with baseline methods, we conducted a user study focusing on four key metrics: lip sync, style consistency, facial expression consistency, and head pose consistency. For baseline methods that cannot generate head motion, we also fix the head of our method during rendering. All comparisons were conducted using a pairwise comparison method, with the motions generated by our method and the competing baseline methods displayed side by side (shuffled in order) and provided with ground truth video as a user reference. After watching the videos, users subjectively selected the results they thought was better. The proportion of user selections was then calculated to quantify

Table 4: Ablation results on CapTalkingHead dataset.

| Method | LVE ↓ | MHD ↓ | FFD ↓ | LODD ↓ | HPDD ↓ |
|---|---|---|---|---|---|
| Clip Text Encoder | 6.57 | 1.81 | 25.15 | 58.62 | 8.24 |
| No Caption | 7.49 | 2.08 | 29.59 | 86.48 | 9.30 |
| Emotion (Audio) Caption Only | 7.39 | 1.96 | 26.18 | 65.82 | 8.62 |
| Style (Video) Caption Only | 6.98 | 1.90 | 23.73 | 62.86 | 8.14 |
| CapTalk (Ours) | 6.44 | 1.80 | 25.14 | 58.27 | 7.59 |

satisfaction. As shown in Table 3, our method significantly outperforms the baseline methods in lip sync, style consistency, and expression and pose consistency. In particular, the style of our generated motions is closer to the ground truth than DiffPoseTalk (Sun et al., 2024) and ARTalk (Chu et al., 2025).

### 5.5 ABLATION STUDY

**Text Encoder.** To verify the effect of text encoders on style text injection, we also tried Clip (Radford et al., 2021) text encoder. The results shown in Table 4 show that t5 (Raffel et al., 2020a) has slightly better control performance.

**Style and Emotion Caption.** To assess the contribution of style and emotion text captions, we conducted ablation experiments by selectively removing these inputs. When all text inputs were removed and the model relied solely on audio, as shown in Table 4, the system was still able to generate synchronized lip movements. However, control over facial expressions and head movements was lost, leading to substantial declines in FFD, LODD, and HPDD metrics. Further, we evaluated the impact of removing style and emotion captions individually. Excluding the style caption resulted in a more pronounced degradation in performance metrics compared to removing the emotion caption. This indicates that style descriptions—particularly those related to mouth opening and head movement amplitude—play a more significant role in controlling expressive aspects of the generated motion. In contrast, emotion text alone contributed less to overall performance, which also explains why our method only achieves results comparable to the baseline on the MEAD dataset when only emotion labels are provided as input. We attribute this to the fact that the model can infer some emotional cues directly from the audio input, whereas style captions captured in the video provides additional details that are not readily available from audio alone.

## 6 DISCUSSION AND CONCLUSION

In this paper, we introduce CapTalk, a novel framework that allows users to directly control the speaking style and emotion of generated motions through textual descriptions. We will also release a corresponding large-scale 3D facial motion dataset with rich speaking style and emotion annotations to facilitate future research in this area. Our experimental results demonstrate that CapTalk outperforms state-of-the-art baseline models in lip synchronization, naturalness of expression, and style control. We believe that the strong generalization and convenient control capabilities of CapTalk make it a promising solution for a wide range of applications, including virtual avatars, language training, and animation for games and films.

**Limitations and future work.** While CapTalk demonstrates strong lip synchronization and effective style control, it has several notable limitations. First, the window-based modeling approach hinders the generation of motion in a fully streamlined and continuous manner. Additionally, the model's limited semantic understanding restricts its ability to produce culturally specific or context-sensitive facial motions. We believe that our work provides a solid foundation for future research in 3D talking head generation. In particular, future efforts will focus on overcoming these challenges, with an emphasis on integrating richer semantic information to enable more context-aware and culturally adaptive head motion synthesis.

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

## A    ETHICS STATEMENT

The advancement of our method in generating photorealistic talking head sequences brings significant opportunities for creative and educational applications, but also introduces ethical challenges, particularly regarding the potential for misuse in deceptive or harmful contexts. We acknowledge the dual-use nature of this technology and the responsibility that comes with its development and dissemination. To promote responsible use, we are committed to transparency in both our research and the outputs generated by our system. We recommend that all synthetic content produced using our method be accompanied by explicit disclosures, such as watermarks or tags, to help users and viewers distinguish between real and generated media. Furthermore, we support the establishment and adoption of industry-wide ethical guidelines and best practices for the creation and distribution of synthetic media. We also recognize the importance of ongoing vigilance and adaptation as the technology evolves. To this end, we will actively engage with the broader community—including ethicists, policymakers, and technologists—to monitor emerging risks and to contribute to the development of effective safeguards. By fostering a culture of ethical awareness and accountability, we aim to maximize the societal benefits of our work while minimizing the potential for harm.

## B    REPRODUCIBILITY STATEMENT

We ensure the reproducibility of this study by describing the details in detail and providing the core code of our multi-scale binary spherical quantization codec and generative model in the supplementary material. The model architecture and training details are presented in this section, and we plan to open source the model code in the future. Data processing is described in the appendix C.

We train our multi-scale encoder-decoder model to obtain binary motion codes, and the code dimension is 32 bits. We use a temporal window size is 100 frames (4 seconds) following (Sun et al., 2024; Chu et al., 2025), and the length of multi-scale code $C_i$ are [1, 5, 25, 50, 100]. This means that the first level code $C_1$ has a shape of $1 \times 32$ and represents part of the information of the entire window (motion length 100). The second level has a code $C_2$ has a shape of $5 \times 32$, and the last level has a code shape of $100 \times 32$. During this stage, we use the AdamW (Loshchilov & Hutter, 2019) optimizer with a learning rate of 1.0e-4, a total batch size of 64, and 100,000 iterations. In the second stage, we train the autoregressive model using the AdamW optimizer, with a learning rate of 1.0e-4, a total batch size of 64, and 100,000 iterations. For the audio encoder, we use the frozen pre-trained Wav2vec 2.0 (Baevski et al., 2020). For the text encoder, we use the frozen pre-trained T5 encoder (Raffel et al., 2020b). During training, we flip the binary codes with a probability of 0.1 and discard the previous actions, style text captions, or emotion text captions with a probability of 0.1 to make the model more robust. All drop probabilities are sampled independently. All training is conducted on an NVIDIA Tesla A100 GPU, taking approximately 28 GPU hours in total (8 hours for the first stage and 20 hours for the second stage).

## C    DATASET

Our dataset is primarily derived from the TalkingHead1KH (Wang et al., 2021) dataset, which comprises approximately 1,000 hours of raw YouTube videos released under the Creative Commons License. To ensure higher quality and longer-duration clips, we implemented a multi-stage preprocessing pipeline. First, we detected, tracked, and cropped face sequences exceeding 8 seconds in length from the original videos. Next, we employed SyncNet (Chung & Zisserman, 2016) to verify audio-visual synchronization, discarding clips with poor lip-audio alignment. We then extracted FLAME (Li et al., 2017) parameters using a hybrid model based on MICA (Zielonka et al., 2022) and EMOCA (Danecek et al., 2022). Finally, we applied the aforementioned Vision-Language Model (VLM) and Audio-Language Model (ALM) annotation procedures to each video segment. After processing, our dataset comprises 24,441 video clips, totaling approximately 200 hours of footage. All videos are standardized to 25 frames per second and 16,000 Hz audio, resulting in 18,074,445 frames, with an average clip duration of approximately 29 seconds. Each video clip is paired with a style description derived from the video content, an emotion annotation extracted from the audio, and corresponding FLAME motion parameters. This makes our dataset the largest dataset with both 3D motion annotations and style text annotations. Some key comparisons are shown in the Table 5.

Table 5: Comparison across existing talking head datasets.

| Dataset | 3D Annotation | Style | Emotion | Hours |
|---|---|---|---|---|
| VOCASET (Cudeiro et al., 2019) | FLAME Mesh | ✘ | ✘ | 0.5 |
| RAVDESS (Livingstone & Russo, 2018) | ✘ | ✘ | ✔ | 1.5 |
| MEAD (Wang et al., 2020) | ✘ | ✘ | ✔ | 3.48 |
| TalkingHead1KH (Wang et al., 2021) | ✘ | ✘ | ✘ | 200 |
| FaMoS (Bolkart et al., 2023) | FLAME Mesh | ✘ | ✘ | 2.7 |
| TFHP (Sun et al., 2024) | FLAME | ✘ | ✘ | 20 |
| MMHead (Wu et al., 2024b) | FLAME | ✘ | ✔ | 49 |
| Express4D (Aloni et al., 2025) | ARKit | ✔ | ✔ | 1.5 |
| **CapTalkingHead (ours)** | FLAME | ✔ | ✔ | 200.8 |

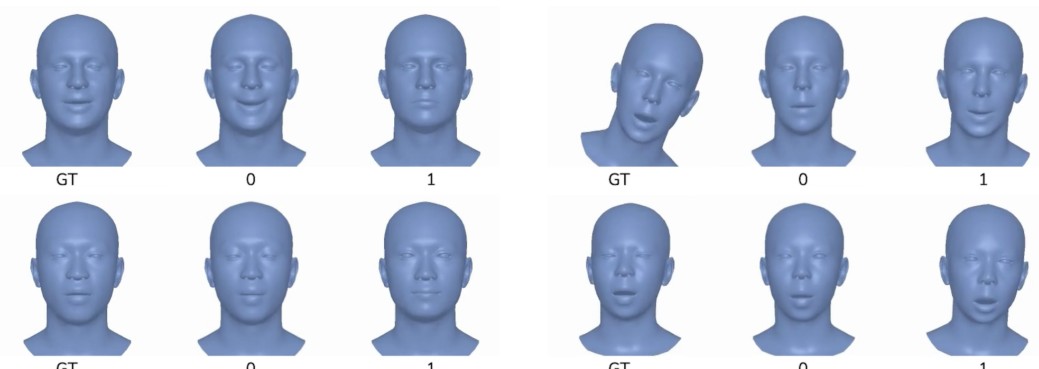

Figure 6: Samples of our user study. Side-by-side videos include ground truth, video 0, and video 1. One of the videos (0 or 1) is generated by our method, and the other is generated by the baseline method, with their order randomized.

## D PRELIMINARIES OF 3DMM

We leverage the FLAME model (Li et al., 2017), a widely adopted 3D morphable model (3DMM) renowned for its geometric accuracy, realistic rendering capabilities, and versatility. FLAME extends beyond static facial models by incorporating parametric controls for identity, pose, and expression, making it suitable for applications such as facial animation, avatar creation, and facial recognition.

In our framework, FLAME serves as the representation for facial motion. We construct a multi-scale codebook using FLAME parameters and learn speech-driven autoregression within this codebook, effectively leveraging the strong geometric priors embedded in FLAME. This approach offers two key advantages: (1) it reduces the high-dimensional complexity of directly modeling mesh vertices, and (2) it enables seamless integration with downstream tasks that utilize FLAME-based representations (Chu et al., 2024; Deng et al., 2024; Chu & Harada, 2024; Xu et al., 2024).

The FLAME model represents the head shape as follows:

$$TP(\hat{\beta}, \hat{\theta}, \hat{\psi}) = \bar{T} + BS(\hat{\beta}; S) + BP(\hat{\theta}; P) + BE(\hat{\psi}; E), \tag{2}$$

where $\bar{T}$ is a template head avatar mesh, $BS(\hat{\beta}; S)$ is the shape blend-shape function accounting for identity-related variation, $BP(\hat{\theta}; P)$ is a corrective pose blend-shape for jaw and neck deformations, and $BE(\hat{\psi}; E)$ captures facial expressions such as eye closure and smiling.

## E  User Study Details

We collected a total of 13 survey responses, with each participant completing 132 questions corresponding to 38 pairwise comparison trials. Of these, 20 comparisons were conducted between our method and baseline methods with fixed head pose, while the remaining 18 comparisons involved our method and baselines with dynamic head pose (DiffPoseTalk and ARTalk). To mitigate potential selection bias, the presentation order of our method and the baseline was randomized in each trial. Specifically, for each comparison, one of the two videos (labeled as 0 or 1) was generated by our method, with the assignment randomized to ensure fairness.

For each comparison, users were asked three or four questions to evaluate the quality of lip sync, overall style consistency and naturalness, expression consistency, and head pose consistency. The questions asked were as follows: Which lip sync is better? Which style is more consistent with the ground truth and natural? Which expression looks more consistent with the ground truth? Which head gesture looks more consistent with the ground truth? Each question was single-choice, requiring users to select video 0 or video 1. For baselines that cannot generate head gestures, only the first three questions were asked. Figure 6 shows some samples from our user study.

## F  Captions in Figure 5

Caption A: {"style caption": "The head movements are subtle, with occasional nods and slight tilts. The mouth movements are natural and expressive, indicating active speech. The facial expressions are neutral to slightly animated, suggesting a conversational tone.", "emotion caption": "Neutral"}

Caption B: {"style caption": "The person in the video appears to be speaking with a positive emotional tone. The head movements are varied, with the person turning their head to the side and then back to the camera, indicating a change in direction or focus. The mouth movements are expressive, with the person opening their mouth wide, suggesting emphasis or a strong point being made. The speaking style seems to be natural and conversational.", "emotion caption": "Happy"}.

Caption C: {"style caption": "The person in the video appears to be speaking with a neutral to slightly serious emotional tone. The head movements are varied, with some moments of nodding and others where the head is tilted slightly. The speaking style is natural, with clear articulation and a moderate pace. The mouth movements are subtle, indicating a controlled and deliberate speech pattern.", "emotion caption": "Happy"}.

Caption D: {"style caption": "The person in the video appears to be speaking with a serious and focused expression. Their head movements are moderate, indicating they are actively engaged in their speech. The mouth movements are subtle, suggesting a controlled and deliberate speaking style. The emotional tone seems to be serious and professional, appropriate for a formal setting like the World Economic Forum. The overall expression is one of confidence and authority.", "emotion caption": "Happy"}.

Caption E: {"style caption": "The person in the video appears to be engaged in a serious conversation on the phone. Their facial expression is one of concern or frustration, with furrowed brows and a slightly open mouth, indicating they might be speaking with urgency or intensity. The head movements are minimal, with slight nods and turns to follow the direction of the conversation. The speaking style is assertive, with a clear and somewhat forceful tone. The mouth movements are exaggerated, with the lips parting slightly more than usual to emphasize the words being spoken. The overall emotional tone is serious and possibly angry or concerned.", "emotion caption": "Angry"}.

## G  The Use of Large Language Models

During the writing process, we utilized Large Language Models (LLM) to assist with grammar checking. The LLM was not used for research ideation or conceptual development. All LLM-checked text was carefully reviewed by the authors to ensure that no changes were made to the intended meaning.

