# OpenReview forum: "CapTalk: Text-Guided Stylization and Speech-Driven 3D Head Animation"
_ICLR.cc/2026/Conference — ICLR 2026 Conference Withdrawn Submission_

### Official Review · Reviewer_tbiV · 2025-10-21

**Soundness:** 2
**Presentation:** 2
**Contribution:** 2
**Rating:** 2
**Confidence:** 4

**Summary:**

The paper proposes CapTalk, a facial animation generation method that can control speaking style with text. The paper also propose a dataset for the generation task.

**Strengths:**

1. Facial animation generation is a promising task.
2. Generating desired speaking style is also promsing.

**Weaknesses:**

1. The paper lacks novelty. Media2face has already achieved speaking style control by text and proposed a large-scale facial motion dataset with speaking styles. CapTalk and the dataset are not the first in the area, and the paper does not discuss Media2face paper.
2. The performance improvement shown in demo is incremental.
3. The paper lacks comparisons with emotional methods, such as emotalk.

**Questions:**

please refer to weakness.

---

### Official Review · Reviewer_bfVu · 2025-10-27

**Soundness:** 2
**Presentation:** 3
**Contribution:** 1
**Rating:** 2
**Confidence:** 5

**Summary:**

The authors propose CapTalk, a text-guided stylization model for 3D talking head generation. To this end, they collected a 200-hour dataset and used multimodal large language models to annotate both style and emotion captions. The model is Transformer-based and generates motion codes in an autoregressive manner. Textual captions are injected through cross-attention to achieve style control.

**Strengths:**

1. A large-scale 200-hour 3D talking head dataset with style and emotion annotations was collected.

2. The proposed CapTalk model achieves state-of-the-art performance in terms of lip synchronization, expression naturalness, and style controllability.

**Weaknesses:**

1. (Major) The model design is an incremental extension of ARTalk, with modifications in motion codec quantization and in how control conditions are injected into the Transformer. In terms of style control, related works such as Media2Face and InstructAvatar have already demonstrated natural-language-guided speaking style control. The authors could better highlight the novelty of the multi-scale quantization codec or provide an ablation on caption injection design to strengthen their technical contribution.

2. (Major) The supplementary video (5.mp4) does not convincingly demonstrate the claimed control ability:
	- B: “Turning their head to the side and then back to the camera” — this behavior is not observed.
	- C: “Head movements are varied” — not more varied than D, which is described as “moderate.”
	- D: “Head movements are moderate; mouth movements are subtle” — actually appears expressive.
	- E: “Furrowed brows; turns to follow” — these are not present.

In summary, the model seems to lack accurate control over head motion amplitude, and the style captions do not clearly correspond to the generated expressions. I suggest adding an experiment that quantifies head pose variance for different style prompts (e.g., subtle, minimal, moderate, varied), and visualizing the distribution with histograms. Since audio also contains cues related to head motion, the same audio segment should be used for fair comparison.

3. (Major) In line 892, the captions in Fig. 5 (C and D) mention “neutral” and “serious” styles, while the corresponding emotion captions are “happy.” It is unclear whether such conflicts arise from the dataset annotations. This inconsistency could make it difficult for the model to effectively use the control conditions (e.g., in Cafe-Talk). How do the authors ensure that emotional cues from audio, the style caption, and the emotion caption remain consistent? Moreover, was any cross-validation performed between the two caption types to mitigate hallucinations from Qwen-based annotation models?

4. (Minor) The abstract claims real-time performance, but no implementation details (e.g., device type or frame rate) are provided to support this claim.

References

- [1] Media2Face: Co-speech facial animation generation with multi-modality guidance, SIGGRAPH'25

- [2] InstructAvatar: Text-guided emotion and motion control for avatar generation, AAAI'25

- [3] Cafe-Talk: Generating 3D talking face animation with multimodal coarse-and fine-grained control, ICLR'25

**Questions:**

Listed in the weakness section.

---

### Official Review · Reviewer_SNL4 · 2025-10-31

**Soundness:** 2
**Presentation:** 2
**Contribution:** 2
**Rating:** 2
**Confidence:** 4

**Summary:**

The paper proposes CapTalk, a framework for text-guided, speech-driven 3D facial animation.
The core idea is to enable dynamic style and emotion control during inference by conditioning the generation process on textual descriptions. To achieve this, the authors:
	1.	Introduce a large-scale dataset (CapTalkingHead) annotated with textual descriptions of both speaking style and emotion, automatically generated via VLM and ALM
	2.	Design an autoregressive generator operating within and across time windows to ensure temporally continuous.

The goal is to allow text-based manipulation of speaking style (e.g., expressive vs. calm) and emotion (e.g., happy, sad) over time, enabling more controllable and adaptive 3D talking head synthesis.

**Strengths:**

•	Using audio-language models for automatic emotion labeling is a practical and scalable approach for large datasets.
•	The paper aims to address real user-facing challenges, such as controllability and expressiveness, moving toward more flexible text-driven animation tools.
•	The idea of supporting dynamic emotion transitions during inference is useful and underexplored in prior work.

**Weaknesses:**

1. Claims not fully substantiated
  - “applying a fixed style or identity throughout an entire audio segment often leads to facial animations that fail to adapt to the dynamic emotional content of speech” → The paper makes this claim but does not show any visual or quantitative example demonstrating this limitation.
- The paper argues that the method supports real-time performance, but no inference time or FPS results are reported
- Dynamic control: Not shown/discussed in the paper/supplemental. All the comparison with different style, uses a constant emotion
2. The motivation for building “a large-scale dataset annotated with textual descriptions of both style and emotion” is clear, but the paper does not adequately explain how the dataset was constructed, why it is necessary beyond existing ones, or how sequences were subsampled from TalkingHead1KH.
3.  It is unclear how the authors accounted for VLM/ALM misclassification errors or evaluated the quality of the generated annotations.
4. The reasoning behind using a multi-scale binary quantized codec instead of a simple VQ-VAE is not justified or ablated.
5. We conduct a comprehensive evaluation of our method against leading academic baselines across two datasets: CapTalkingHead and MEAD (Wang et al., 2020). The baseline methods FaceFormer (Fan et al., 2022), CodeTalker (Xing et al., 2023), and SelfTalk (Peng et al. 2023a) are mesh-based methods, where we input the corresponding mesh and specify its first speaker identity for inference.” → This comparison seems unfair since the baselines are trained on different datasets.
6. The model takes both Cap style + Cap emotion as input, but there is no analysis of how much influence each component contributes; more detailed ablations are needed.

**Questions:**

1.  How were the subsampled sequences from TalkingHead1KH selected, and what criteria were used?
2.	How did the authors handle VLM/ALM misclassifications or noisy annotations?
3.	What advantages does the multi-scale binary quantized codec provide compared to standard VQ-VAE variants?
4.	Can the authors provide runtime results to substantiate the “real-time” claim?
5.	How does the model’s output differ when using only style versus only emotion captions? Where does the mail style benefit come from ?

---

### Official Review · Reviewer_od69 · 2025-11-02

**Soundness:** 3
**Presentation:** 3
**Contribution:** 3
**Rating:** 6
**Confidence:** 3

**Summary:**

This paper proposes a speech-driven talking head generation framework that enables the user to control the speaking style and emotion via textual descriptions. This allows the generation of synchronized lip movements and facial expressions that reflect the provided descriptions. In addition, a new dataset is introduced, which will be made public. Comparison with SOTA methods on 2 datasets is presented together with an ablation study.

**Strengths:**

- A new large-scale 3D facial motion dataset with annotations for speaking styles and emotions has been collected and will be publicly released.

- SOTA results achieved.

- Controlling h the speaking style and emotion of generated motions via textual descriptions is novel.

**Weaknesses:**

- The user study is rather weak, only 13 subjects participated which makes the results less convincing.

- The videos in the supplementary material do not seem to contain blinks. This is in contrast to other SOTA methods which generate blinks.
Why this happens?

- It would be good if details on how the collected dataset is split into training/validation/test sets are added.

- Table 4, which shows the ablation study, is a bit confusing. It would be better if one component at a time is removed, e.g, only the audio caption is removed or only the video caption removed. If the authors believe there is a reason showing the table in the current form, which shows only what is kept and what is removed, it would be good to explain this.

- There is a typo in the caption of Fig. 1: a framework generate 3D head motions -> a framework to generate 3D head motions

**Questions:**

Please see above.

---

### Note · Authors · 2025-12-08

I have read and agree with the venue's withdrawal policy on behalf of myself and my co-authors.